# Attitudes, beliefs and behaviors of religiosity, spirituality, and cultural competence in the medical profession: A cross-sectional survey study

**Victoria Dillard**[1]\*, **Julia Moss**[2], **Natalie Padgett**[1], **Xiyan Tan**[3], **Ann Blair Kennedy**[1]\*

**1** University of South Carolina School of Medicine Greenville, Greenville, South Carolina, United States of America, **2** Department of Internal Medicine, University of Utah School of Medicine, Salt Lake City, Utah, United States of America, **3** Clemson University School of Mathematical and Statistical Sciences, Clemson, South Carolina, United States of America

\* vdillard@email.sc.edu (VD); kenneda5@greenvillemed.sc.edu (ABK)

## Abstract

### Introduction

Religion and spirituality play important roles in the lives of many, including healthcare providers and their patients. The purpose of this study was to examine the relationships between religion, spirituality, and cultural competence of healthcare providers.

### Methods

Physicians, residents, and medical students were recruited through social platforms to complete an electronically delivered survey, gathering data regarding demographics, cultural competency, religiosity, and spirituality. Four composite variables were created to categorize cultural competency: Patient Care Knowledge, Patient Care Skills/Abilities, Professional Interactions, and Systems Level Interactions. Study participants (n = 144) were grouped as Christian (n = 95)/non-Christian (n = 49) and highly religious (n = 62)/not highly religious (n = 82); each group received a score in the four categories. Wilcoxon rank sum and Chi-square tests were used for analysis of continuous and discrete variables.

### Results

A total of 144 individuals completed the survey with the majority having completed medical school (n = 87), identifying as women (n = 108), white (n = 85), Christian (n = 95), and not highly religious (n = 82). There were no significant differences amongst Christian versus non-Christian groups or highly religious versus not highly religious groups when comparing their patient care knowledge (p = .563, p = .457), skills/abilities (p = .423, p = .51), professional interactions (p = .191, p = .439), or systems level interaction scores (p = .809, p = .078). Nevertheless, participants reported decreased knowledge of different healing traditions (90%) and decreased skills inquiring about religious/spiritual and cultural beliefs that may affect patient care (91% and 88%). Providers also reported rarely referring patients to religious services (86%).

**Data Availability Statement:** The data has been uploaded to the Figshare repository (DOI: 10.6084/m9.figshare.14442560).

**Funding:** The author(s) received no specific funding for this work.

**Competing interests:** The authors have declared that no competing interests exist.

## Conclusions

Although this study demonstrated no significant impact of healthcare providers' religious/spiritual beliefs on the ability to deliver culturally competent care, it did reveal gaps around how religion and spirituality interact with health and healthcare. This suggests a need for improved cultural competence education.

## Introduction

According to the 2014 Religious Landscape Study by the Pew Research Center, 77% of all adults in the United States identify with a religious faith [1]. Various studies have found a link between religious involvement and decreased mortality [2–4], leading many to further examine the role religion and spirituality should play in healthcare delivery.

According to a 2001 article written by the American Psychological Society, religion describes specific behavior patterns and practices that align with a traditional system of faith and worship, where spirituality involves the pursuit of an ultimate truth and reality that is sacred to an individual [5]. It is important to note the differences between these two concepts —as spirituality should not be confined to religion, but appreciated within a broader definition that encompasses meaning, purpose, and relationships [6].

In 2015, the U.S. experienced a drop in traditional religious beliefs and practices and a rise in what is considered the religious "nones": a group of Americans who are not religiously affiliated, but retain their belief in God [1]. This shift in the religious landscape has further widened the generational gap and challenged traditional methods of religious observance. Nevertheless, 77% of all adults continue to identify with a religious faith, and 41% of these persons say their religious beliefs lay the foundation for their moral sense and ethical motives [1]. As religion and spirituality remain a complex and multidimensional phenomena, they continue to add to the depth of diversity in healthcare organizations.

Cultural competence is defined as "a set of congruent behaviors, attitudes, and policies that come together in a system, agency, or among professionals and enable that system, agency, or those professionals to work effectively in cross-cultural situations" [7]. Some of the barriers that affect the ability of patients to access care can be minimized through cultural competence techniques, including communication with patients in their preferred language, leadership and workforce diversity, cross-cultural training of providers, identification of structural barriers, and the provision of culturally appropriate health education materials [8, 9]. Individual providers' attributes, behaviors, and attitudes impact interactions with patients as well as the environment of their workplace and is therefore important to understand as a contributor to the cultural competence of the health system as a whole [9].

When we consider the intersection of religion, spirituality, and culturally competent healthcare, one study found that among a group of neonatologists, there were no differences amongst religious and non-religious providers in the decision to withhold or withdraw life-saving treatment to neonates in the intensive care unit [10]. Nevertheless, 7% of the study participants reported no obligation to refer patients to an alternate provider for options that they would object [10]. This is further highlighted in a study done by Curlin et al., which demonstrated that 18% of U.S. physicians reported no moral obligation to refer a patient to an alternative provider willing to perform a procedure or offer a treatment that they feel is morally wrong [11]. This challenges one to consider how personal religious and spiritual beliefs may play a role in interactions with patients in the healthcare setting. As globalization continues and the

population of the United States becomes increasingly diverse, the ability to care for diverse people is becoming ever more important for healthcare providers. Furthermore, the ability to do this despite one's own personal religious and spiritual belief systems reinforces an important angle of social and behavioral research.

Although robust literature exists that explores provider religiosity and spirituality and the impact this may have on different aspects of culturally competent healthcare (specifically, spiritual care in clinical settings), to our knowledge, there have been no studies that have specifically assessed how providers' religious and spiritual identities impact the domains of cross-cultural knowledge, culturally aware patient care skills and abilities, cross-cultural professional interactions, and healthcare systems-level interactions.

This study examined attitudes, behaviors and beliefs associated with religious and spiritual variables of cultural competence among attending physicians, fellows, resident physicians, and medical students in the United States. The results of this study will be used to guide future research and inform cultural competence training efforts in healthcare and medical education settings.

## Methods

### Study design and inclusion criteria

A cross-sectional survey design was used to gather data regarding participant demographics (race/ethnicity, age, and gender identity), cultural sensitivity, religiosity, and spirituality from medical students in the clinical years, residents, fellows, and attending physicians (either MD or DO) (MsRP) practicing in the United States at the time of the survey. To be included in the survey, participants were required to meet the following qualities: 1) reside, work, and/or attend school in the United States, 2) be a medical student (MD or DO) in the clinical years, a resident, fellow, or a practicing physician (MD or DO), and 3) agree to participate in the survey. An online survey was selected to allow for a broader distribution of the instrument across geographic areas and levels of training. Participants were recruited through social media (Facebook, Twitter) and local and national (i.e. Student National Medical Association) email list-servs. Participation in the study was voluntary and uncompensated. All participants were adults. Written consent to participate in the study was obtained from all participants.

### Survey instrument

**Instrument resources.**   The survey instrument (S1 Appendix) was developed based on the adoption of three primary resources: Clinical Cultural Competency Questionnaire (CCCQ, Post-Training Version), Promoting Cultural and Linguistic Competency Self-Assessment Checklist for Personnel Providing Primary Healthcare Services, and Cultural Competence Self-Assessment Questionnaire (CCSAQ).

The following represent the survey domains of the present study: *demographics; patient population race/ethnicity; patient population religion; participant religiosity and spirituality; the interaction of religion, spirituality, and patient care; knowledge; skills; comfort; communication; diversity of interactions; sociocultural identities;* and *training and education.* The questions that were included in each of these domains were selected by expert faculty in the field of cultural competency and health disparities.

**Demographics & patient population.**   The first part of the survey collected *demographic* data, which included MsRP level of training, state and type of practice/medical school, specialty (intended or actual), race/ethnicity, gender identity, age, and religious identity. Respondents were then asked to provide their best estimates of the demographics of their patient population in the section, *patient population by ethnicity, race, and religion.* Respondents were

required to estimate, and they were offered nine percentage ranges to choose from as well as an option for "do not know." This metric was inspired by the Cultural Competence Self-Assessment Questionnaire (CCSAQ), a tool developed by the Portland Research and Training Center that is now applied in a variety of settings: from child and family service agencies to curriculum planning (the alpha coefficient for all but one subscale in the questionnaire was 0.8) [12].

**Personal values.** Participants were asked about their personal values and beliefs toward religiosity and spirituality (*participant religiosity and spirituality* domain). Definitions used for religion and spirituality are included in the survey instrument in S1 Appendix. This section focused on assessing the providers' engagement with religious and spiritual practices, and they were evaluated with 4, 5, and 6-point Likert scales. MsRP were asked the frequency in which they attend religious services, which was graded on a 4-point Likert scale ("at least once a week," "1–2 times a month," "1–2 times a year," or "never"). Frequency of prayer and practice of mindfulness and/or meditation with or without movement (yoga, Thai Chai, etc.) was graded on a 6-point Likert scale ("multiple times a day," "daily," "weekly," "monthly," or "rarely"). The importance of religion and spirituality to personal life was assessed with a 5-point Likert ("very important," "somewhat important," "I don't know," "not too important," and "not at all important"). These questions were based on the Pew Research Center's Religiosity Landscape Study [1].

**Intersections with patient care.** The *intersection of religion, spirituality, and patient care* domain asked respondents how often they ask patients about religious, spiritual, or cultural beliefs and practices as well as how often they refer their patients to a chaplain or other services to meet their religious needs (Likert 5-point: "never," "rarely," "sometimes," "frequently," or "always"). These questions were influenced by the Pew Research Center's study on Religiosity in the United States [1].

**Knowledge, skills, & comfort.** The *knowledge*, *skills*, and *comfort* domains were influenced by the Clinical Cultural Competency Questionnaire (CCCQ, Post-Training Version) with permission from the survey author, Robert Like, at the Robert Wood Johnson Medical School [13]. The CCCQ is a 63-item measure originally developed by the Rutgers Robert Wood Johnson Medical School to assess physicians' competency in providing high quality care to diverse patient populations. The questionnaire assesses knowledge of health disparities, skills in delivering culturally competent healthcare and dealing with sociocultural issues, comfortability with facing cross-cultural situations, self-awareness of biases and prejudices, and the importance of cultural competency training [13]. The questionnaire has been translated into at least six different languages and used to develop cultural competence educational programs in healthcare. Research has shown the questionnaire to be quite reliable, with an alpha coefficient greater than 0.8 for all subscales; the alpha coefficient was >0.9 for the *knowledge* section, >0.87 for the *skills* section, and >0.8 for the *comfort* section [14]. Each question was answered using a Likert 5-point scale: "never," "rarely," "sometimes," "frequently," "always", or "I don't know."

Some questions were modified to better accommodate the target population (MsRP). In the *knowledge* domain, respondents were asked about health disparities and socioeconomic and sociocultural factors that affect populations in their community. The *skills* section asked respondents to reflect on their ability to deliver culturally competent care; for example, performing and delivering culturally sensitive exams and education. In the *comfort* domain, participants were asked about how comfortable they feel in certain cross-cultural interactions like caring for patients of a different sexual and/or gender identity than themselves or working with healthcare professionals from culturally diverse backgrounds.

**Communication and diversity of interactions.** The *communication* and *diversity of interactions* domains were adopted and modified from the Promoting Cultural and Linguistic

Competency Self-Assessment Checklist for Personnel Providing Primary Healthcare Services tool, which was developed by Georgetown University's National Center for Cultural Competence (NCCC), and it is a widely used self-assessment tool [15]. It is a publicly available tool that assesses physical environment, materials and resources, communication styles, and values and attitudes toward cultural and linguistic competence in health and human service settings at the physician and organizational level. It provides specific examples on the values and practices that help to foster and promote cultural and linguistic competence [15]. As reliability and validity statistics are not available for this tool, we performed a principal components analysis to determine the Cronback's alpha score of the survey and its components (Table 1) [16]. The questionnaire had a high level of internal consistency, as determined by Cronbach's alpha of .878.

Questions in the *communication* section focused on proper communication with patients in their language of preference as well as the display of multimedia (artwork, printed materials, pictures, etc.) to reflect the diverse cultural and ethnic backgrounds they serve. In the *diversity of interactions* section, respondents were asked about their advocacy for policies and procedures that are culturally and linguistically inclusive, their ability to intervene when observing culturally insensitive behaviors, and their participation in professional development training to enhance knowledge and skills in cultural competence. Each question was answered using a Likert 5-point scale: "never," "rarely," "sometimes," "frequently," "always", or "I don't know."

**Table 1. Items for composite variables with Cronbach's alpha score.**

| Composite variable | Items within variable |
|---|---|
| **Patient Care Knowledge (23 dichotomized items; Cronbach's Alpha = .889)** | Describe the percentage of your population by ethnicity and race |
| | • Native American/American Indian or Alaska Native |
| | • Asian |
| | • Black or African American |
| | • Hispanic or Latino/a/x |
| | • Native Hawaiian or Pacific Islander |
| | • White |
| | Describe the percentage of your population by religion |
| | • Agnostic |
| | • Atheist |
| | • Baha'i |
| | • Buddhism |
| | • Catholic |
| | • Hindu |
| | • Jewish |
| | • Mormon |
| | • Muslim |
| | • Protestant |
| | • Sikhism |
| | How knowledgeable are you about |
| | • Demographics of diverse racial, and ethnic groups in my community |
| | • Health disparities affecting the populations in my community |
| | • Socioeconomic factors that impact health |
| | • Socioeconomic factors affecting the populations in my community |
| | • Different healing traditions (e.g. Ayurvedic Medicine, Traditional Chinese Medicine) |
| | • Historical and contemporary impact of racism, bias, prejudice and discrimination in healthcare experienced by various populations |

*(Continued)*

**Table 1.** (Continued)

| Composite variable | Items within variable |
|---|---|
| **Patient Care Skills and Abilities (15 dichotomized items; Cronbach's Alpha = 0.791)** | How religion and spirituality interact with patient care |
| | • How often do you ask your patients about their religious or spiritual beliefs? |
| | • How often do you ask your patients about their cultural beliefs and practices? |
| | • How often do you refer patients to a chaplain or other service to meet their religious needs? |
| | How skilled are you in |
| | • Eliciting the patient's perspective about health and illness? (e.g. its etiology, name, treatment, course, prognosis) |
| | • Performing a culturally sensitive physical examination? |
| | • Providing culturally sensitive patient education, counseling, and treatment plans? |
| | • Apologizing for cross-cultural misunderstandings or errors? |
| | • Treating a patient who makes derogatory comments about your racial or ethnic background? |
| | • Caring for a patient who uses folk healers or alternative therapies |
| | How comfortable do you feel about the cross-cultural interactions |
| | • Caring for patients from culturally diverse backgrounds |
| | • Caring for patients with limited English proficiency |
| | • Caring for patients of a different sexual orientation than you |
| | • Caring for patients of a different gender identity than you |
| | • Interpreting different cultural expressions of pain, distress, and suffering |
| | • Advising a patient to change behaviors or practices related to cultural beliefs that impair one's health |
| **Professional Interactions (4 dichotomized items; Cronbach's Alpha = 0.497)** | How comfortable do you feel about the cross-cultural interactions |
| | • Working with a colleague who makes derogatory remarks about patients from a particular ethnic group? |
| | • Working with health care professionals from culturally diverse backgrounds |
| | How often are the following true for you in your medical practice? |
| | • I intervene when I observe other students, staff, or clients within my program or agency engaging in behaviors that show cultural insensitivity, racial biases, and prejudice. |
| | • I have participated in professional development and training to enhance my knowledge and skills in the provision of services to culturally, and linguistically diverse groups. |
| **Systems Level Interactions (4 dichotomized items; Cronbach's Alpha = 0.662)** | How often are the following true for you in your medical practice? |
| | • Artwork, printed materials, pictures, videos, and health education materials that I use and/or display in my work environment reflect the different cultures and ethnic backgrounds of the clients I serve. |
| | • I ensure that my patients have access to healthcare services in the language they prefer. |
| | • I ensure that all notices and communications to individuals and families are written in their language of origin and take into account the average literacy levels of those that I serve. |
| | • I advocate for the review of my program's or agency's mission statement, goals, policies, and procedures to ensure that they incorporate principles and practices that promote cultural and linguistic competence. |

Cronbach's Alpha on combined composite variables (46 items) = 0.878.

**Sociocultural identities.** In the *sociocultural identities* section, adapted from the CCCQ, MsRP were asked about how important they believe sociocultural identities are in workplace interactions with patients, staff, and colleagues in addition to their awareness of their own biases and prejudices toward certain races, ethnicities, and cultures. Each question was answered using a Likert 4-point scale: "not at all important," "not too important," "somewhat important," or "very important."

**Training and education.** Lastly, the *training and education* portion of this survey inquired about hours of cultural competence training at different levels of medical practice.

**Survey validation.** The survey instrument was previewed by six experienced physicians and two medical students to provide stakeholders' feedback on the question stems, responses, design, and flow of the questionnaire.

## Procedures and statistical analysis

The survey was distributed electronically to participants via email and social media from May to August 2019. Demographic data were compiled both for the entire sample and for four separate groups, Christian versus Non-Christian and highly religious (HR) versus not highly religious (NHR), for further analysis. It is important to note that the structure of this analysis was based on the representation of Christianity as the largest religious group worldwide.

Participants that identified as "Christian" were placed in one group. All other religious identities were placed in the "Non-Christian" group. Furthermore, Highly Religious (HR) and Not Highly Religious (NHR) groups were created based on responses to frequency in attending religious services, frequency with praying, and importance of religion. Respondents were placed in the HR group if they 1) attend religious services "at least once a week" or "1–2 times a month," 2) pray "multiple times a day" or "daily," and 3) chose "very important" or "somewhat important" to the importance of religion to personal life question. Respondents were placed in the NHR group if they 1) attend religious services "1–2 times a year" or "never," 2) pray "weekly," "monthly," "rarely," or "never," and 3) chose "not too important," "not at all important," or "I don't know" to the importance of religion to personal life question. As this study was seen as a preliminary exploration of differences between groups, sample size calculations were not performed prior to analysis; however, post-hoc power analyses were conducted for any non-significant findings.

**Composite variables.** In an effort to demonstrate levels of cultural competency, the question items were grouped into four composite variables: *Patient Care Knowledge*, *Patient Care Skills and Abilities*, *Professional Interactions*, and *Systems Level Interactions*. All questions were dichotomized. For the Likert response questions, the top two responses (e.g. "extremely" and "quite a bit") received a "1" and the three other responses (e.g. "not at all," "a little," or "somewhat") received a "0." For the questions regarding the percentage of their patient population by race/ethnicity and religion, respondents were scored a "1" for choosing any percentage range for race/ethnicity and religious population makeup, and those that chose "do not know" were scored a "0." S2 Appendix illustrates the procedure flowchart. Table 1 provides the items used to construct the four composite variables along with the Cronbach's alpha score for each composite variable.

**Statistical analysis.** Quantitative data were analyzed using R statistical software (R Foundation for Statistical Computing, version 3.6.1) with alpha set at 0.05. Continuous variables are reported as mean ± standard deviation. Discrete variables are reported as N (%). Continuous variables were analyzed using Wilcoxon rank-sum test or two independent samples t-test which is two-tailed, while discrete variables were analyzed using Chi-square. Post-hoc power analyses were conducted using G*Power 3.1 software for any non-significant results.

**Ethical approval.** The University of South Carolina Institutional Review Board approved this project (Pro00087751). Participants were given a description of the survey and asked to "opt in" if they were willing to participate.

## Results

### Participants

214 participants began the survey. 72 participants had incomplete responses, leaving 156 completed responses. 12 of the 156 had to be removed due to not fitting the inclusion criteria questions, leading to a total of 144 participants. The majority completed medical school (n = 87) and self-identified as female (n = 108), white (n = 85), Christian (n = 95), and not highly religious (n = 82). Characteristics of the 144 participants who completed the survey are shown in Table 2 and S3 Appendix.

**Table 2. Participant information.**

| N | | 144 |
|---|---|---|
| Age, Mean ± SD | | 34.4 ± 10.5 |
| | Median (IQR) | 31 (26, 40.2) |
| Position, N(%) | | |
| | Medical Student | 57 (39.58) |
| | Resident | 23 (15.97) |
| | Fellow | 5 (3.47) |
| | Attending Physician | 59 (40.97) |
| Location of Practice, N(%)[a] | | |
| | Urban | 85 (59.02) |
| | Suburban | 55 (38.19) |
| | Rural | 14 (9.72) |
| | Tribal | 0 (0) |
| Race/Ethnicity, N(%)[b] | | |
| | American Indian or Alaska Native | 3 (2.08) |
| | Asian | 7 (4.86) |
| | Black or African American | 43 (29.86) |
| | Hispanic or Latino | 6 (4.17) |
| | Native Hawaiian or Pacific Islander | 0 (0) |
| | White | 85 (59.03) |
| | Other | 3 (2.08) |
| | Decline to Answer | 3 (2.08) |
| Gender Identity, N(%) | | |
| | Woman | 108 (75) |
| | Man | 34 (23.61) |
| | Genderqueer | 0 (0) |
| | Decline to Answer | 2 (1.39) |
| Religious Identity, N(%) | | |
| | Christian | 95 (65.97) |
| | Non-Christian | 49 (34.03) |
| Level of Religiosity, N (%) | | |
| | Highly Religious | 62 (43.06) |
| | Not Highly Religious | 82 (56.94) |

[a] Participants were allowed to select more than one practice setting.

[b] Participants were allowed to select more than one race/ethnic identity.

## Composite variables

**Patient care knowledge.** The *Patient Care Knowledge* variable assessed participants' knowledge of their patient population's race/ethnicity and religious affiliations (Fig 1). Furthermore, it measured their awareness of health disparities, biases and prejudices, as well as socioeconomic and sociocultural factors that may affect populations in their community. With a potential maximum score of 23, the median score for Christians was 13 compared to 15 for non-Christian groups (there was no statistical difference between these two groups, p = 0.563). Moreover, the median score for the HR group was 13.5 and 15 for the NHR group (there was no statistical difference between these groups, p = 0.457).

**Patient care skills and abilities.** The *Patient Care Skills and Abilities* variable focused on how providers incorporate religious, spiritual, and cultural beliefs and practices into clinical

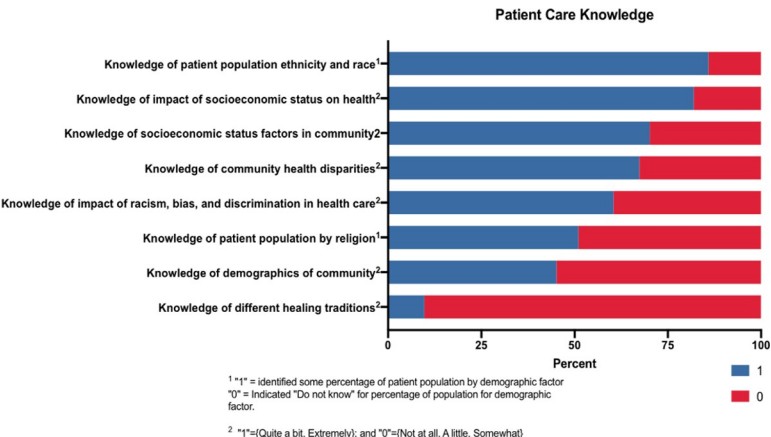

**Fig 1. Patient care knowledge results.** [1] "1" = Identified some percentage of patient population by demographic factor "0" = Indicated "Do not know" for percentage of population for demographic factor. [2] "1" = {Quite a bit, Extremely}; and "0" = {Not at all, A little, Somewhat}.

experiences (Fig 2). With a potential maximum score of 15, the median score for each of the groups (Christian, non-Christian, HR, NHR) was 7 (Christian versus non-Christian, p = 0.423; HR vs NHR, p = 0.51).

**Professional interactions.** The *Professional Interactions* variable asked participants to respond to situational questions that centered on intervening when observing cultural insensitivity, racial bias, and prejudice (Fig 3). Furthermore, it inquired about provider comfortability with working with healthcare professionals of diverse cultural backgrounds as well as any previous professional development training that has helped to enhance cultural competence skills. With a potential maximum score of four, the median score for each of the groups (Christian, non-Christian, HR, NHR) was two (Christian versus non-Christian, p = 0.191; HR vs NHR, p = 0.439).

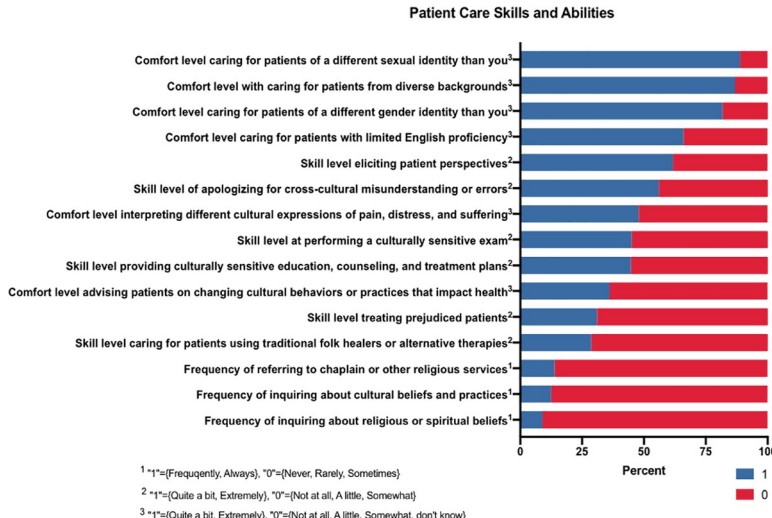

**Fig 2. Patient care skills and abilities results.** [1] "1" = {Frequently, Always}, "0" = {Never, Rarely, Sometimes}. [2] "1" = {Quite a bit, Extremely}, "0" = {Not at all, A little, Somewhat}. [3] "1" = {Quite a bit, Extremely}, "0" = {Not at all, A little, Somewhat, don't know}.

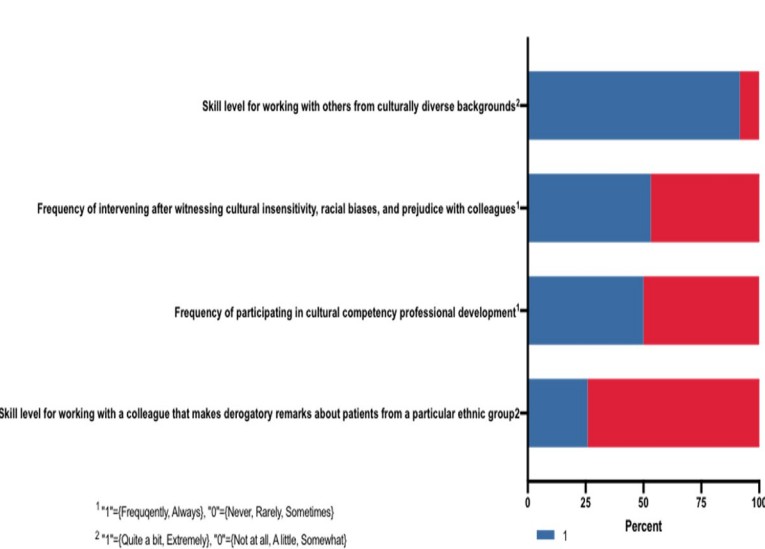

**Fig 3. Professional interactions results.** [1] "1" = {Frequently, Always}, "0" = {Never, Rarely, Sometimes}. [2] "1" = {Quite a bit, Extremely}, "0" = {Not at all, A little, Somewhat}.

**Systems level interactions.** The *Systems Level Interactions* variable focused on culturally sensitive communications within the workplace as well as advocacy at the level of the institutional mission, policies, and procedures that promote cultural and linguistic competence (Fig 4). With a potential maximum score of four, the median score for each of the groups (Christian, non-Christian, HR, NHR) was two (Christian vs. non-Christian, p = 0.809; HR vs. NHR, p = 0.078).

**Score comparison.** There were no significant differences between groups when comparing their scores for each of the composite variables (Table 3). Additionally, each group had a potential maximum composite score of 46 with the combination of all variables. Christians had an overall median score of 25, while non-Christians had an overall score of 26 (there was no statistical difference between the groups, p = 0.72). Both HR and NHR groups had a median

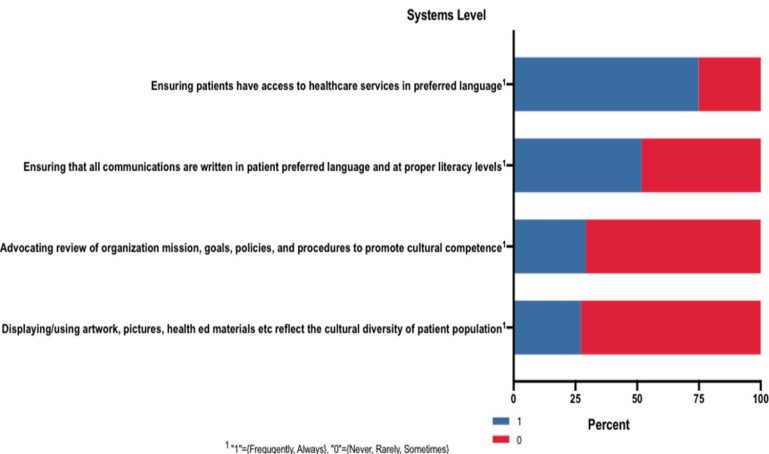

**Fig 4. Systems level interactions results.** [1] "1" = {Frequently, Always}, "0" = {Never, Rarely, Sometimes}.

**Table 3. Demographic data and total score comparisons for each of the groups: Christian versus non-christian and HR versus NHR.**

| | | Christian | | | High Religious | | |
|---|---|---|---|---|---|---|---|
| | | Christian | Not Christ | p-value | High Religious | Not Highly | p-value |
| **N** | | **95** | **49** | | **62** | **82** | |
| Age, Mean ± SD | | 33.3 ± 10.1 | 36.5 ± 11.3 | | 33.7 ± 10.9 | 35 ± 10.3 | |
| | Median (IQR) | 30 (26, 39) | 36 (27, 43) | | 30 (25.2, 40) | 32 (26, 41.8) | |
| Position, N(%) | | | | | | | |
| | Medical Student | 37 (38.95) | 20 (40.82) | | 24 (38.71) | 33 (40.24) | |
| | Resident | 18 (18.95) | 5 (10.20) | | 13 (20.97) | 10 (12.20) | |
| | Fellow | 4 (4.21) | 1 (2.04) | | 3 (4.84) | 2 (2.44) | |
| | Attending Physician | 36 (37.89) | 23 (46.94) | | 22 (35.48) | 37 (45.12) | |
| Location of Practice, N(%) | | | | | | | |
| | Urban | 58 (61.05) | 27 (55.10) | | 38 (61.29) | 47 (57.32) | |
| | Suburban | 31 (32.63) | 24 (48.98) | | 18 (29.03) | 37 (45.12) | |
| | Rural | 11 (11.58) | 3 (6.12) | | 10 (16.13) | 4 (4.88) | |
| | Tribal | 0 (0) | 0 (0) | | 0 (0) | 0 (0) | |
| Race/Ethnicity, N(%) | | | | | | | |
| | American Indian or Alaska Native | 3 (3.15) | 0 (0) | | 3 (4.84) | 0 (0) | |
| | Asian | 4 (4.21) | 3 (6.12) | | 3 (4.84) | 4 (4.88) | |
| | Black or African American | 30 (31.58) | 13 (26.53) | | 22 (35.48) | 21 (25.61) | |
| | Hispanic or Latino | 4 (4.21) | 2 (4.08) | | 2 (3.23) | 4 (4.88) | |
| | Native Hawaiian or Pacific Islander | 0 (0) | 0 (0) | | 0 (0) | 0 (0) | |
| | White | 57 (60) | 28 (57.14) | | 34 (54.84) | 51 (62.20) | |
| | Other | 1 (1.05) | 2 (4.08) | | 0 (0) | 3 (3.66) | |
| | Decline to Answer | 0 (0) | 3 (6.12) | | 0 (0) | 3 (3.66) | |
| Gender Identity, N(%) | | | | | | | |
| | Woman | 72 (75.79) | 36 (73.47) | | 43 (69.35) | 65 (79.27) | |
| | Man | 23 (24.21) | 11 (22.45) | | 19 (30.65) | 15 (18.29) | |
| PC Knowledge Score, Mean ± SD (total: 23) | | 13.9 ± 5.65 | 14.6 ± 5.2 | | 13.7 ± 5.71 | 14.4 ± 5.34 | |
| | Median (IQR) | 13 (9.5) | 15 (10) | 0.563 | 13.5 (10) | 15 (9.75) | 0.457 |
| PC Skills/Abilities Score, Mean ± SD (total: 15) | | 6.92 ± 3.38 | 7.37 ± 3.09 | | 6.87 ± 3.25 | 7.22 ± 3.31 | |
| | Median (IQR) | 7 (5.5) | 7 (5) | 0.423 | 7 (4) | 7 (5) | 0.51 |
| Professional Interactions Score, Mean ± SD (total: 4) | | 2.27 ± 1.13 | 2.06 ± 1.03 | | 2.27 ± 1.13 | 2.14 ± 1.08 | |
| | Median (IQR) | 2 (2) | 2 (2) | 0.191 | 2 (1) | 2 (2) | 0.439 |
| Systems Level Interactions Score, Mean ± SD (total: 4) | | 1.83 ± 1.23 | 1.8 ± 1.43 | | 1.6 ± 1.27 | 1.99 ± 1.30 | |
| | Median (IQR) | 2 (2) | 2 (2) | 0.809 | 2 (2.5) | 2 (2) | 0.078 |
| | Overall Score, Mean ± SD (total: 46) | 24.90 ± 8.47 | 25.80 ± 6.82 | 0.52 | 24.45 ± 8.02 | 25.77 ± 7.87 | 0.326 |
| | Median (IQR) | 25 (10) | 26 (8) | 0.72 | 25 (10) | 25 (9.75) | 0.551 |

P-values of continuous variables are from Wilcox.test; for categorical variables, P-values are from Chi square test.

overall score of 25 (there was no statistical difference between the groups, p = 0.551). The null results in the comparison between groups led us to conduct a post hoc power analysis to determine if our study was adequately powered. The priori analysis indicated that a power (1 - β) set at 0.80 and α = .05, two-tailed, sample sizes are suggested to include 67 participants in each group to detect medium effects (d = .5); large effects (d = .8) could be detected with only 27 participants in each group. However, the post hoc analysis to determine achieved power based upon means and standard deviations for each group yielded an effect size d = 0.117 with Power = 0.099.

## Discussion

This cross-sectional survey study was conducted to better understand how health care providers' attitudes, behaviors, and beliefs toward religion and spirituality impact the delivery of culturally competent care. While our study found no significant differences between Christian versus Non-Christians and Highly Religious versus Not Highly Religious groups when analyzing their knowledge of patient populations, culturally competent skills and abilities, professionalism, and health care systems level engagement, specific sample-wide deficiencies were discovered. An examination of the individual composite variables used to conduct this analysis revealed deficiencies in incorporating patient religious/spiritual beliefs into clinical practice, in dealing with derogatory remarks toward patients or colleagues of diverse backgrounds in the clinical setting, and in promoting systems level changes to reflect the diverse workforce and patient populations served. These findings are further discussed below.

### Patient care knowledge

Respondents reported their patient population by ethnicity, race, and religion as well as ranked their level of knowledge on various sociocultural and socioeconomic factors (health disparities, different healing traditions, bias, prejudice, and historical impacts of racism). Respondents were not asked to describe their patient population by sexual orientation or gender identity. Although there was no statistical difference between Christian versus Non-Christian groups or HR versus NHR groups on their *Patient Care Knowledge* scores, closer inspection of the individual components that comprise this composite variable demonstrate disparities across all respondents in regards to knowledge of their patient populations.

More than 80% of the participants indicated they had a substantial amount of knowledge about how socioeconomic factors impact health, and 70% indicated they understand how those socioeconomic factors impact their specific populations. Yet, only 9% of the respondents indicated that they were knowledgeable about different healing traditions. This suggests a need for more rigorous education on non-Western medicine and complementary and alternative methods of healing, and it challenges providers to broaden their perspectives on medicine and healing to be more inclusive of non-traditional modalities [17]. Understanding these various methods of healing is important to better serve diverse racial, ethnic, cultural, and religious populations in our communities.

Less than half of the participants indicated they were knowledgeable about the racial and ethnic groups in their communities. When analyzing healthcare providers' abilities to describe their patient populations by ethnicity and race, they perceived "White," "Black or African American," and "Hispanic or Latino" to make up the majority of their respective patient populations. Nevertheless, over 25% of respondents were unable to estimate the number of Native Americans, Native Hawaiians, or Pacific Islanders in their communities as evidenced by their selection of "do not know" for those racial and ethnic categories. This may be indicative of low representation of these groups in healthcare providers' respective communities, but it may also suggest a need for more training on how to respectfully inquire about a patient's racial/ethnic and cultural background. Though these groups represent a relatively small percentage of the U.S. population, it is important to recognize and honor these differences between patients as they may help to more effectively deliver care to these diverse populations.

In regards to describing patient population by religion, participants perceived their patient populations to be primarily Protestant (69%), Catholic (67%), and Muslim (62%). The three religions with the lowest response rate (respondents who were unable to answer) were Buddhism, Baha'i, and Sikhism. This, interestingly, correlates to the religious identity of the survey participants in that zero participants identified with Buddhism, Baha'i, or Sikhism, whereas

the majority of participants identified as Protestant (52%) or Catholic (14%). This suggests a need for more awareness on how one's personal religious beliefs may be different than that of the patient and how this may impact their health and healthcare. Although the majority religion in America comprises Christian faiths, more education on other religions may expand knowledge and enhance the doctor-patient relationship. One study found that 80% of patients reported physicians never or rarely offer opportunities to discuss religious and spiritual issues [18]. This aligns with our findings when we consider how respondents were largely unable to estimate the minority religious makeup of their patient populations. This suggests a need for more support and training for these types of discussions in the healthcare setting.

## Patient care skills & abilities

The ability to deliver a level of care that addresses the unique needs of patients from diverse cultural and religious backgrounds is a key skill set needed to ensure equitable healthcare for all. Though this particular section demonstrated no significant differences between the skills and abilities of Christian versus non-Christian faiths and HR versus NHR groups, an examination of the individual components that comprise this composite variable reveals gaps across all respondents in how religion and spirituality interact with patient care.

The data suggests that holistically, our respondents are not inquiring about religious and cultural practices and beliefs that may affect patient care. Over 90% of respondents indicated that they "never," "rarely," or "sometimes" ask patients about their religious or spiritual beliefs. Simultaneously, over 80% of respondents indicated that they "never," "rarely," or "sometimes" ask patients about their cultural beliefs and practices. Our findings are in alignment with a systematic review of religion and spirituality discussions with patients, finding that these factors are infrequently addressed but increase with terminal illness and high provider religiosity and spirituality [19]. Possible barriers to discussing spirituality include insufficient time and training, confusion over differences between religion and spirituality, lack of vocabulary, invasion of privacy, and lack of clarity around the physician role in these discussions [19]. Nevertheless, religious, spiritual, and cultural beliefs and practices are often interwoven, thus, cultural competency requires a thorough understanding of the patient's religion and spirituality.

Furthermore, there appears to be decreased ability to care for patients who use folk healers and alternative therapies with greater than 70% of respondents indicating that they are "not at all," "a little," or "somewhat" skilled in this area. These results align with a 2005 study on resident preparedness, which found that residents reported a lack of preparedness to treat patients with the following: religious beliefs or practices that don't align with traditional Western medicine, religious beliefs that affect treatment, and alternative/complementary medicine preferences [20]. This lack of improvement in these skill sets calls for a review of cultural and religious competence curriculum and training in both undergraduate and graduate medical education. In a 2001 article on spirituality and medical practice, The American Academy of Family Physicians encouraged the use of the HOPE Questionnaire, a tool to help integrate a spiritual assessment into the medical interview [21]. The adoption of a resource like this would support students, residents, and attending physicians in having these important but often overlooked conversations. Furthermore, the tool may help to foster a stronger spiritual self-understanding for providers themselves, yielding a more unbiased, patient-focused, and nonjudgmental spiritual assessment of patients.

In contrast to the above findings, 87% of our study participants reported high levels of comfortability with caring for patients from culturally diverse backgrounds. This gap in comfort versus skill suggests a need for more granular solutions like the HOPE Questionnaire discussed previously. Further, increasing knowledge and awareness around the process of

referring patients to chaplains and locations of worship centers and meditation spaces is essential. Although literature suggests doctors prefer chaplains to facilitate these discussions, there appear to be gaps in rates of physicians recommending chaplain referral and actually making the referral. Our study revealed that 86% of study participants reported not referring their patients to a chaplain or other service, which calls to attention the importance of expanding the concepts of medical management and intervention in the healthcare setting to encompass religious and spiritual needs.

This variable also explores respondents' ability to provide care to patients who make derogatory remarks about their racial and ethnic backgrounds. Over 69% of respondents noted their ability to treat patients was diminished when facing these overt biases. In a 2019 study on *Physician and Trainee Experiences with Patient Bias*, researchers found that these types of patient behaviors lead to anger, emotional pain, confusion, and fear, which can be distracting for the provider [22]. Lack of skills, support, and guidance from attendings and institutions as well as fear of fracturing the therapeutic alliance were noted as the primary barriers to responding in such situations [22]. In accordance with the present study, this further underscores the need for more training on how to appropriately navigate situations with biased patients and highlights the need for clear institutional policies and medical curricula that support and respect the increasingly diverse medical workforce.

Over half of the participants reported low skill levels in performing culturally sensitive physical exams, providing culturally sensitive patient education and treatment plans, interpreting different cultural expressions of pain, and advising patients to change cultural behaviors or practices that impair one's health. These vast sets of skills are imperative to ensuring positive patient outcomes, compliance, and trust of the healthcare system. Further development of cultural sensitivity programs should hone in on these important competencies which were found to be lacking in the majority of this study's current MsRP.

Our results in this particular section appear to align with several international studies on spirituality and spiritual care in the clinical setting. Specifically, in Taiwan, Australia, and New Zealand, there is a call for spiritual care education amongst both physicians and nurses in order to meet the growing needs of patients [19, 23, 24]. Although our study took a broader approach to analyzing cultural competence, this alignment suggests a global need for enhanced cultural competency curriculum across multiple disciplines.

### Professional interactions

Employee conduct and behavior is a reflection of an institution's culture and its ability to promote professionalism across the organization. Our study included four questions on professional interactions in the workplace to assess its overlap with concepts of culture and inclusivity.

Over 74% of our respondents reported discomfort when working with colleagues that make derogatory remarks about patients. We see a similar response in our skills and abilities subsection, where providers indicated discomfort with patients who make derogatory remarks about providers. Nevertheless, just over half (53%) of our study participants reported intervening when they see culturally insensitive acts, racial biases, and prejudice, suggesting that although providers do not feel comfortable, many do feel a moral pull to advocate in situations that they deem unjust. For the half of our respondents that indicated they do not intervene in these situations, this raises the question on how institutions are equipping learners and attendings on how to handle difficult patient-doctor and/or doctor-doctor interactions.

Nearly 40% of respondents were medical students, for whom there may be even more hesitation in speaking out against unfair situations, especially in settings where they are being

evaluated. Nevertheless, there is power in training students on how to approach advocacy, as these efforts would positively impact patient care and empower future generations of providers.

This section also reveals that only half of our participants indicated they "frequently" or "always" participate in cultural competency training. Our respondents noted that medical school was the primary setting in which they received this training (66%) with engagement in residency and Continuing Medical Education (CME) dropping off significantly (28% residency, 21% CME). According to a 2013 study on cultural competence training in Graduate Medical Education (GME), there appears to be varying requirements across specialties and fragmented implementation of cultural sensitivity education, which may help to explain the findings in our study [25]. Further, another study, which focused on cultural competence training in U.S. medical schools, found that of the 18 medical programs reviewed, 67% mandated this type of education [26]. With such contrast between different levels of medical training, further study on how to standardize best practices across medical schools and residency programs may help with continued engagement throughout undergraduate and graduate medical education.

### Systems level interactions

Health care systems play a broader role in caring for patients. Messaging and other actions taken at this level can affect many people–both the patient population as well as healthcare providers and other staff. Scores for this category were between 1.6 and 1.99 out of 4, indicating that less than half of the respondents were taking action to improve cultural sensitivity at the systems level. However, there was significant variation among the four questions in this section.

Interestingly, the item in this category with the least respondents reporting it was "always" or "frequently" true for their practice was regarding whether artwork or health education materials reflected the diverse populations they serve (27%). Relatively few respondents (29%) reported advocating for review of their institution's mission statement, policies, procedures, or goals to ensure they are culturally competent. Just over half of respondents reported that they ensure materials for patients are written in their preferred language and at an appropriate literacy level. 75% of respondents report they ensure their patients have access to care in their preferred language; this item achieved the highest score in this category and represents an action that can be completed at an individual versus systems level, which may explain the greater engagement. This higher level of engagement may also be due to system-level policies dictating the use of interpreters for patients whose primary language is not English (i.e. the Affordable Care Act of 2010), suggesting that the creation of policies regarding cultural and linguistic competence can improve provider engagement with system level efforts.

The sample population may be one factor influencing the responses for this category. Systems-level change is often left to those at the highest level of power within an institution. Those still in training formed 60% of the sample. Some of the questions in this category lend themselves more to attendings, as they require the ability to make changes in the workplace. Further study of the ability to make systems-level changes from various levels of training would be beneficial.

### Limitations

Identifying the limitations of this study provides proper context on which it should be interpreted. Although 214 participants attempted the survey, only 144 completed the entire instrument and met our inclusion criteria. Only completed surveys were analyzed which limited the

power of the study. Further, the majority of the participants identified as white, female, Christian, and not highly religious. Although effort was made to recruit a diverse group of participants, the aforementioned demographics were the overwhelming majority. Additionally, the recruitment platforms present their own limitations (Facebook, Twitter, and email listservs). The lack of diversity in the respondents may be a reflection of the social networks through which the survey was disseminated and regional bias. Different results may emerge with a more diverse sample. Moreover, while a sample size of 54 total individuals would be required to see large effect sizes, the post hoc analysis yielded a power of ~0.1. Thus, we cannot rule out that there may be a difference between groups if sample sizes were larger.

Questions on religiosity and spirituality in particular were based on frequency of prayer and attending religious services. We recognize that there are various modes of religious practice, and these more structured analyses may not reflect this diversity. Our study is further limited by the absence of a validated tool like the DUREL (The Duke University Religious Index) to measure religious involvement [27]. Additionally, the survey may have created a self-reporting bias of participants. There was no instrument to validate responses to each question. Lastly, the survey was one-sided–failing to ask patient/client health outcomes and perspectives could potentially bias the data set. In order to improve client/patient health outcomes through increased cultural competence, measures of both provider and patient outcomes must be considered [28].

### Future implications

Future studies should focus on how specialty practice, level of training, gender, and race/ethnicity intersect with religion and cultural competency. Interventions to improve the incorporation of religious and cultural values in the clinical encounter and care plan are also needed, and an analysis on how this impacts patient satisfaction and outcomes would help to fill gaps in the cultural competency literature.

### Conclusion

Analyzing healthcare providers' attributes, behaviors, and attitudes toward culture, religion, and spirituality is important because these factors impact interactions with patients as well as the environment of the workplace. Although this study revealed no differences in the ability to deliver culturally competent care between religious or non-religious healthcare providers, it unveiled lower levels of knowledge and skill (across all groups) in incorporating diverse religious and cultural beliefs into clinical practice, in dealing with derogatory remarks toward patients or colleagues of diverse backgrounds in the clinical setting, and in promoting systems-level changes to reflect the diverse workforce and patient populations served. These results demonstrate the need for improved and sustained cultural competency education across all levels of medical training.

### Supporting information

**S1 Appendix. Survey instrument.**
(PDF)

**S2 Appendix. Procedure flowchart.**
(TIF)

**S3 Appendix. Number of survey participants by state.** Survey disseminated May-August 2019. Darker colors indicate greater numbers of survey participants.
(TIF)

## Acknowledgments

We would like to thank the UofSC School of Medicine for their support in executing this project.

## Author Contributions

**Conceptualization:** Julia Moss, Natalie Padgett, Ann Blair Kennedy.

**Data curation:** Xiyan Tan.

**Formal analysis:** Xiyan Tan, Ann Blair Kennedy.

**Investigation:** Victoria Dillard, Julia Moss, Natalie Padgett, Ann Blair Kennedy.

**Methodology:** Victoria Dillard, Julia Moss, Natalie Padgett, Ann Blair Kennedy.

**Project administration:** Victoria Dillard, Julia Moss, Ann Blair Kennedy.

**Resources:** Xiyan Tan.

**Supervision:** Ann Blair Kennedy.

**Validation:** Ann Blair Kennedy.

**Visualization:** Victoria Dillard, Natalie Padgett, Ann Blair Kennedy.

**Writing – original draft:** Victoria Dillard, Julia Moss, Natalie Padgett, Ann Blair Kennedy.

**Writing – review & editing:** Victoria Dillard, Julia Moss, Natalie Padgett, Ann Blair Kennedy.

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
