## [Decision Letter · Decision Letter 0]

22 Dec 2020

PONE-D-20-23707

Attitudes, beliefs and behaviors of religiosity, spirituality, and cultural competence in the medical profession: A cross-sectional survey study

PLOS ONE

Dear Ms.Dillard, 

Thank you for submitting your manuscript to PLOS ONE. After careful consideration, we feel that it has merit but does not fully meet PLOS ONE’s publication criteria as it currently stands. Therefore, we invite you to submit a revised version of the manuscript that addresses the points raised during the review process.

Please address the contextual and methodological issues raised by the reviewers.

We look forward to receiving your revised manuscript.

Kind regards,

Rosemary Frey

Academic Editor

PLOS ONE

Journal Requirements:

2. Please refer to any sample size calculations performed prior to participant recruitment. If these were not performed please justify the reasons. Please refer to our statistical reporting guidelines for assistance (https://journals.plos.org/plosone/s/submission-guidelines.#loc-statistical-reporting)

3. Thank you for including your ethics statement: 

"The xxxxxx Institutional Review Board approved this project (PRO#xxxxx). Participants were given a description of the survey and asked to “opt in” if they were willing to participate".   

4. Please provide additional details regarding participant consent.

In the ethics statement in the Methods and online submission information, please ensure that you have specified what type you obtained (for instance, written or verbal, and if verbal, how it was documented and witnessed).

If your study included minors, state whether you obtained consent from parents or guardians.

If the need for consent was waived by the ethics committee, please include this information.

6. We note that Figure 2 in your submission contains map images which may be copyrighted.

We require you to either (a) present written permission from the copyright holder to publish these figure specifically under the CC BY 4.0 license, or (b) remove the figure from your submission:

b. If you are unable to obtain permission from the original copyright holder to publish these figure under the CC BY 4.0 license or if the copyright holder’s requirements are incompatible with the CC BY 4.0 license, please either i) remove the figure or ii) supply a replacement figure that complies with the CC BY 4.0 license. Please check copyright information on all replacement figures and update the figure caption with source information. If applicable, please specify in the figure caption text when a figure is similar but not identical to the original image and is therefore for illustrative purposes only.

Reviewers' comments:

Reviewer's Responses to Questions

**Comments to the Author**

1. Is the manuscript technically sound, and do the data support the conclusions?

Reviewer #1: Yes

Reviewer #2: Yes

2. Has the statistical analysis been performed appropriately and rigorously? 

Reviewer #1: Yes

Reviewer #2: Yes

3. Have the authors made all data underlying the findings in their manuscript fully available?

Reviewer #1: Yes

Reviewer #2: Yes

4. Is the manuscript presented in an intelligible fashion and written in standard English?

Reviewer #1: Yes

Reviewer #2: Yes

5. Review Comments to the Author

Reviewer #1: Thank you for the invitation to review this article.

Background:

Lines 82-90: Here you are describing how different religions hold different attitudes on same sex marriages – and then discuss immigration where an increase of immigrants has had a negative impact, but it’s not clear what the negative impact has been upon. (Apologies if I have missed this).

The connection of this sentence to the need to deliver culturally appropriate care is not clearly demonstrated in the current text, as you explain what culturally competent care is after this sentence when it should come earlier. You work would also benefit from a sentence or two discussing how religion and spirituality are an important component of culturally competent care. The background would benefit from discussing how religious and spiritual beliefs often intertwine with cultural beliefs, and so true cultural competency requires an understanding of the patient’s religion and spirituality.

There are a few reviews, key textbooks that are worth mentioning all which discuss the role of a healthcare professionals religious and spiritual beliefs and the influence these can have during clinical encounters.

It is well-established that if a healthcare professional does not have a religious or spiritual belief delivering spiritual care is difficult often awkward, different beliefs in relation to the patient can also be a barrier as they wish to not offend the patient. It is important to acknowledge that much of this work has been conducted on nursing professionals, specific diseases and healthcare conditions (e.g., cancer, palliative care, pain), if you are referring to a lack of evidence in specific medical situations then please clarify this.

I have recommended some texts worth adding to this section:

Shah, S., Frey, R., Shipman, K., Gardiner, F., & Milne, H. (2018). A survey to explore health care staff perceptions of spirituality and spiritual care working in a single district health area in New Zealand. European Journal of Integrative Medicine, 22, 1-9.

Chen, M. L., Chen, Y. H., Lin, L. C., & Chuang, L. L. (2020). Factors influencing the self‐perceived competencies in spiritual care of nurses in the long‐term care facilities. Journal of Nursing Management.

Delbridge, E., Taylor, J., & Hanson, C. (2014). Honoring the “spiritual” in biopsychosocial-spiritual health care: Medical family therapists on the front lines of graduate education, clinical practice, and research. In Medical family therapy (pp. 197-216). Springer, Cham

Donohue, P. K., Boss, R. D., Aucott, S. W., Keene, E. A., & Teague, P. (2010). The impact of neonatologists' religiosity and spirituality on health care delivery for high-risk neonates. Journal of palliative medicine, 13(10), 1219-1224.

Austin, P., Macleod, R., Siddall, P., McSherry, W., & Egan, R. (2017). Spiritual care training is needed for clinical and non-clinical staff to manage patients’ spiritual needs. Journal for the Study of Spirituality, 7(1), 50-63.

Best, M., Butow, P., & Olver, I. (2016). Why do we find it so hard to discuss spirituality? A qualitative exploration of attitudinal barriers. Journal of clinical medicine, 5(9), 77.

Best, M., Butow, P., & Olver, I. (2016). Doctors discussing religion and spirituality: A systematic literature review. Palliative Medicine, 30(4), 327-337.

The uniqueness of the research is not clear from the background – there have been plenty of studies that have explored health care professionals/physicians attitudes behaviours and beliefs in relation to religion and spirituality and the role these have in delivering spiritual care/culturally competent care. Equally there have been several reviews that have identified that cultural needs of patients are not addressed within clinical consultations with physicians, psychologists - we already have a wealth of studies exploring and identifying what’s missing – what gaps will your current study address having a clear aim and concluding statement about specific gaps you wish to address will benefit the background.

Methods:

This section was very difficult to follow, may I suggest breaking it down in to the subheadings of the each of the scales that were used to build your survey, and under each sub-heading discussing the measure psychometrics and which items/domains you took for this survey.

So something like:

We developed our instrument on three questionnaires that have been used in healthcare service research the process will be described below:

CCCQ –

What the measure is, and how it used previously (with supporting references), what are its psychometrics, which questions you used in your survey, which items you amended and whether the psychometrics were impacted by the changes you made.

Follow the same format for each of the questionnaires (cultural linguistic checklist, CCSAQ) you used to build you survey then finish with how the survey was peer-reviewed by six experienced physicians and two medical students to provide stakeholders’ feedback on the question stems, responses, design, and flow of the questionnaire.

How were participants identified? What sampling methods did you use? You used social media – which platforms did you use (e.g., twitter, facebook Instagram Linkedin?).

Was a sample size calculated – as you are assessing a lot of variables was this conducted when designing the original statistical analysis plan? If not why was this not conducted?

Results:

Table 2 – not all of the data presented tallies up to N=144 (Location of practice, gender of the population, race/ethincity) this can be confusing to the reader – I am assuming the location of practice reflects that participants practiced at various locations, can this please be made clearer in the table layout.

You haven’t discussed any regressions analyses, consider removing this from the analysis section, or add a section to the work to say that further exploratory analyses were not completed due to non-significant results

Qualitative data collection was not mentioned in the methods, and but appear in the results but very little is added to the findings, either add a section around how qualitative data was also collected at the end of the survey through open-ended questions or remove this bit. If you intend to keep this bit, then please add the analysis you did (e.g., descriptive thematic analysis) and what were the main findings.

Discussion:

Similar comments are noted to the results, there are several studies that can be used to compare findings to; please consider using some of the refs from earlier to discuss findings from your study.

Truong, M., Paradies, Y., & Priest, N. (2014). Interventions to improve cultural competency in healthcare: a systematic review of reviews. BMC health services research, 14(1), 99.

This is an umbrella review making reference to a lot of useful papers that are worth considering in your discussion.

There is limited discussion of the limitations – very little recommendations can be made from your paper as the sample is too homogeneous, the response rate was moderate – but as there is limited discussion of recruitment strategies, its hard to determine whether this was to be expected or not. Please consider adding a sentence or two on this.

Reviewer #2: The manuscript is written and structured well, but has some areas that need improvement.

1. The main challenge with this study is the lack of statistically significant results, hence relying on indications from inferential statistics. This needs to be strengthen in the discussion to make up for the lack of the former.

2. The abstract, under methods, needs more information - e.g. number of participants, analysis.]

3. The introduction needs improvement. There is abundant literature on cultural competency and cultural humility in healthcare, and currently the text merely refers to little.

Also, on page 3, the authors suggest that spirituality only emerged recently due to the shifting demographics presented. However, this is inaccurate and further literature on spirituality as a concept and spiritually-sensitive practice will help strengthen the contextual information of the study.

4. Equally, there is more literature regarding how clinicians take into account religion and spirituality when making decisions in practice. I would recommend to the authors to explore and re-position the context of the study.

5. A justification of why the self-administered online survey was the best tool to meet the aims of the study would be helpful in appreciating the strengths of the method.

6. More information about the validity of the CCCQ and self-assessment checklist from the NCCC is necessary.

7. The study divided participants in the four groups mentioned, but no significant results were produced. Were there other divisions attempted which may had been more telling?

8. On page 17 (first line), the text reads, '...were asked to describe...'. Yet, participants did not describe anything in their responses but a Likert style survey tool was used.

9. Instead of 'Buddhist', the religion, 'Buddhism' should be used in the text.

10. It is unclear whether the researchers asked about sexuality or sex in the demographics. On pages 19-20, this is further confusing.

11. At the end of the discussion, further analysis alluding to the transferability of the findings may be useful, especially as to make this relatable to other contexts and an international audience.

12. In the limitations, the authors do not talk about the lack of statistically significant results.

13. Figure 1 does not add anything but is redundant to the text that explains the process.

14. Figure 2 can be substituted with a couple of statements in the text - methods - that talk about participants.

Once again, this is an interesting study but needs to take into account abundant work undertaken in the field, to make a stronger claim for adding to the literature.

6. PLOS authors have the option to publish the peer review history of their article (what does this mean?). If published, this will include your full peer review and any attached files.

Reviewer #1: No

Reviewer #2: No

---

## [Author Response · Author response to Decision Letter 0]

5 Feb 2021

Dear Reviewers,

Thank you for the feedback on our manuscript.

We have included a document entitled, "Response to Reviewers," with detailed information on how we addressed each of your comments.

Thank you so much for the opportunity.

Best,

Victoria Dillard, MS

Ann Blair Kennedy, LMT, BCTMB, DrPH

---

## [Decision Letter · Decision Letter 1]

16 Mar 2021

PONE-D-20-23707R1

Attitudes, beliefs and behaviors of religiosity, spirituality, and cultural competence in the medical profession: A cross-sectional survey study

PLOS ONE

Dear Ms Dillard,

Thank you for submitting your manuscript to PLOS ONE. After careful consideration, we feel that it has merit but does not fully meet PLOS ONE’s publication criteria as it currently stands. Therefore, we invite you to submit a revised version of the manuscript that addresses the points raised during the review process.

Please address the methodological issues raised by reviewer one as these are of greatest concern.

We look forward to receiving your revised manuscript.

Kind regards,

Rosemary Frey

Academic Editor

PLOS ONE

Reviewers' comments:

Reviewer's Responses to Questions

**Comments to the Author**

1. If the authors have adequately addressed your comments raised in a previous round of review and you feel that this manuscript is now acceptable for publication, you may indicate that here to bypass the “Comments to the Author” section, enter your conflict of interest statement in the “Confidential to Editor” section, and submit your "Accept" recommendation.

Reviewer #1: (No Response)

Reviewer #2: All comments have been addressed

2. Is the manuscript technically sound, and do the data support the conclusions?

Reviewer #1: Partly

Reviewer #2: Yes

3. Has the statistical analysis been performed appropriately and rigorously? 

Reviewer #1: Yes

Reviewer #2: Yes

4. Have the authors made all data underlying the findings in their manuscript fully available?

Reviewer #1: Yes

Reviewer #2: Yes

5. Is the manuscript presented in an intelligible fashion and written in standard English?

Reviewer #1: Yes

Reviewer #2: Yes

6. Review Comments to the Author

Reviewer #1: Thank you for resubmitting this.

Thank you for suggesting the changes, but I am afraid the rationale for the study is still not quite clear or contextualised in the evidence-base.

There are many aspects to culturally competent care as you have identified, why have you chosen specifically on religion/spirituality? Why not look at cultural knowledge? As religion and spirituality is not the only aspect of cross-cultural care? What about clinicians overcoming language barriers, cultural beliefs that are not related to religion/spirituality?

I will reiterate that the novelty of your study is still not clear as there are ample reviews around delivering and supporting implementation of culturally competent care in various clinical contexts (e.g., cancer, mental health), which has looked at systems level approach also (i.e., clinician characteristics as barriers/facilitators, organisations as barriers/facilitators, clinical training barriers/facilitators and patient characteristics). How is your study different there is a clear enough rationale why exploring physicians RS would impact culturally appropriate care - why has this been given importance in delivering culturally competent care? The two studies you cite are not enough to support this as culturally competent care is more than patients religious and spiritual beliefs.

In addition you have created a measure for exploring personal values (this was exploring intrinsic and extrinsic religiosity) there are validated measures for exploring these variables you measured e.g., DUREL for extrinsic religiosity, Religious orientation scale Intrinsic religiosity im not entirely sure why this was the case? this further limits the validity of your findings as this survey item has not been tested for validity and reliability.

I'm not entirely sure why the analysis was ran in R now instead of SPSS? Have you re-ran the analysis?

Reviewer #2: Many thanks for submitting the revised manuscript. The re-structuring, clarity of information and enhancement of content have certainly improved the submission. I would like to offer that some further proofreading will help tidy up very few typos. While the cross-reference to the Figure 2 (in the text) is deleted (as this is moved to supplementary documents), the sentence still ends with 'within' but is incomplete (no page number in the revised document). I would also encourage the authors to triple check all references matching the list at the end and citation style. In the text, the name Austin appears, for example, as an in-text citation, instead of the number assigned to the reference in the list at the end.

7. PLOS authors have the option to publish the peer review history of their article (what does this mean?). If published, this will include your full peer review and any attached files.

Reviewer #1: No

Reviewer #2: No

---

## [Author Response · Author response to Decision Letter 1]

16 Apr 2021

Dear Reviewers,

Thank you for the opportunity to revise our manuscript entitled: Attitudes, beliefs and behaviors of religiosity, spirituality, and cultural competence in the medical profession: A cross-sectional survey study.

We have carefully reviewed your comments and hope that the additional context around the rationale for our study is acceptable. Please find our responses in the "Response to Reviewers" document attached.

Thank You,

Victoria Dillard, MS

Ann Blair Kennedy, LMT, BCTMB, DrPH

---

## [Decision Letter · Decision Letter 2]

24 May 2021

Attitudes, beliefs and behaviors of religiosity, spirituality, and cultural competence in the medical profession: A cross-sectional survey study

PONE-D-20-23707R2

Dear Ms. Dillard,

We’re pleased to inform you that your manuscript has been judged scientifically suitable for publication and will be formally accepted for publication once it meets all outstanding technical requirements.

Kind regards,

Rosemary Frey

Academic Editor

PLOS ONE

Additional Editor Comments (optional):

Reviewers' comments:

Reviewer's Responses to Questions

**Comments to the Author**

1. If the authors have adequately addressed your comments raised in a previous round of review and you feel that this manuscript is now acceptable for publication, you may indicate that here to bypass the “Comments to the Author” section, enter your conflict of interest statement in the “Confidential to Editor” section, and submit your "Accept" recommendation.

Reviewer #1: All comments have been addressed

Reviewer #2: All comments have been addressed

2. Is the manuscript technically sound, and do the data support the conclusions?

Reviewer #1: Yes

Reviewer #2: Yes

3. Has the statistical analysis been performed appropriately and rigorously? 

Reviewer #1: Yes

Reviewer #2: Yes

4. Have the authors made all data underlying the findings in their manuscript fully available?

Reviewer #1: Yes

Reviewer #2: Yes

5. Is the manuscript presented in an intelligible fashion and written in standard English?

Reviewer #1: Yes

Reviewer #2: Yes

6. Review Comments to the Author

Reviewer #1: (No Response)

Reviewer #2: Thank you for submitting the revised version of the paper. All comments have been addressed, and I have no further feedback to offer.

7. PLOS authors have the option to publish the peer review history of their article (what does this mean?). If published, this will include your full peer review and any attached files.

Reviewer #1: No

Reviewer #2: No

---

## [Editor Report · Acceptance letter]

1 Jun 2021

PONE-D-20-23707R2 

Attitudes, beliefs and behaviors of religiosity, spirituality, and cultural competence in the medical profession: A cross-sectional survey study 

Dear Dr. Dillard:

I'm pleased to inform you that your manuscript has been deemed suitable for publication in PLOS ONE. Congratulations! Your manuscript is now with our production department. 

Kind regards, 

on behalf of

Dr. Rosemary Frey 

Academic Editor

PLOS ONE